# Perma Curette and Hysteroscopy: An Observational Study About Endometrial Sampling

**DOI:** 10.3390/biomedicines13051113

**Published:** 2025-05-04

**Authors:** Carmen Imma Aquino, Daniela Surico, Francesca Miglino, Arianna Ligori, Daniela Ferrante, Valentino Remorgida

**Affiliations:** 1Department of Translational Medicine, University of Piemonte Orientale, Gynecology and Obstetrics, ‘Maggiore della Carità’ Hospital, 28100 Novara, Italy; c.immaquino@gmail.com (C.I.A.); 20042770@studenti.uniupo.it (F.M.); 20027942@studenti.uniupo.it (A.L.); valentino.remorgida@uniupo.it (V.R.); 2Medical Statistics, Department of Translational Medicine, University of Piemonte Orientale, 28100 Novara, Italy; daniela.ferrante@med.uniupo.it

**Keywords:** Perma curette, hysteroscopy, endometrial sampling, endometrial cancer

## Abstract

The role of blind endometrial sampling, in the era of hysteroscopy-guided biopsy, can be only considered as a screening tool or a first-line approach if a hysteroscopy cannot be performed for whatever reason. Several devices are available, with Perma (a sharp-edged spatula sliding inside a flexible cannula) being one of them. **Objectives**: The aims of this study were to compare the concordance of blind to visual endometrial sampling, and the influence of operators’ experience on the results. **Materials and Methods**: Women undergoing hysteroscopy were invited to undergo a Perma biopsy as well. If accepted, a Perma sampling was performed before the hysteroscopy and only if there was no cervical dilatation (as an office setting). The operator was randomly chosen between expert (two staff members) and non-expert (two residents) operators. All cases were collected at the AOU Maggiore della Carità, Novara, Italy. Categorical variables were presented in number and percentage (%) and continuous variables were presented as mean ± SD. The association between categorical variables was evaluated using Fisher’s exact test. Clinical outcomes were analyzed, and the results were first compared within the same patient and subsequent within the doctors’ group (inter- and intra-variation) in terms of Cohen’s Kappa. **Results**: 82 women performed both hysteroscopy and Perma. A sensitivity of 81.8% and specificity of 100% was found when Perma was compared to hysteroscopy (the gold standard). The comparison between valid vs. invalid samples in terms of sufficient evaluable tissue was not significant (*p* = 0.583). There are no statistical associations with body mass index, parity, or previous intrauterine surgery related to the outcomes of hysteroscopy and Perma. Cohen’s Kappa between non-experts was 0.43 (moderate), between experts was 0.30 (fair), with the highest concordance being between one non- and one expert (0.68 = substantial). Perma represents a pragmatic diagnostic tool, which could also be used in outpatient.

## 1. Introduction

Hysteroscopy is commonly used in the diagnosis and treatment of uterine pathologies such as endometrial hyperplasia, polyps, and endometrial carcinoma [1]. Bozzini is considered the father of endoscopy since, in 1807, he was the first to create a device capable of inspecting some cavities of the human body [2]. Desormeaux subsequently invented the hysteroscope, while Pantaleoni was the first to see clearly inside the uterine cavity. Dupley and Clado used an endoscope that consisted of an open tube with a battery-powered light source, but blood and mucus frequently impeded vision [2]. In the following decades, a system of the rinsing and distension of the cavities with fluid was developed. The diameters of the instrument were shortened and reduced, leading to modern hysteroscopes made up of rigid and thin tubes, with a diameter of a few millimeters, and equipped with optical fibers [2]. The classical distinction between diagnostic or operative hysteroscopy nowadays almost completely disappeared as technological improvements has pushed the limits of the so called “office hysteroscopy”, and this technique is the gold standard [1,2,3,4].

With office hysteroscopy, intrauterine diseases can be detected and treated in the exact same setting (“see and treat approach”). As the ‘see-and-treat’ approach brought the benefits of inpatient surgery to the office, hysteroscopic surgical advancements fundamentally transformed the approach to treating intrauterine diseases [5]. Larger abnormalities are frequently addressed in the operating room, while in-office hysteroscopy was primarily focused on small pathology as a supplement to its diagnosis. Diagnostic hysteroscopy is still a useful technique for direct endometrial sampling [5] and can be the first line of treatment for some gynecological conditions, even though the role of other imaging tools in the accurate assessment of benign uterine diseases is expanding, particularly beyond direct hysteroscopic visualization. Today’s pre-operative examination of uterine pathology is based on contemporary ultrasound analysis, and office hysteroscopy has growing significance as a careful approach for more complicated diagnosis and larger lesions. Hysteroscopy is now recognized as a viable, economical, and useful treatment option for nearly all intrauterine conditions [5].

One of the most frequent hysteroscopic procedures, i.e., performed in case of irregular uterine bleeding and infertility, is endometrial sampling. Nevertheless, both guided and blind endometrial sampling techniques are available [6,7]. The sensitivity and specificity of hysteroscopic-guided endometrial biopsy are 100% and 72.8–91%, respectively. Several blind devices for endometrial sampling are available such as Novak curette, Vabra/pipelle aspirator, and endo-sampler. Sensitivity and specificity ranges vary: 57% and 97% (Novak curette), 88.2% and 88.7% (Vabra), 60–100% and 51.4–100% (pipelle aspirator), 93% and 100% (endo-sampler) [6,7,8,9,10]. Pipelle is a transparent and flexible polypropylene sheath with a blunt and rounded distal extremity; a negative pressure is created inside the endometrial cavity by pulling on the internal piston [3]. Novak, on the other hand, is a stainless-steel curette with an external diameter of 5 mm, serrated edges and distal opening, while Vabra is an aspirator with a maximum diameter of 2.5 mm [11]. Endo-sampler is a valid method of endometrial sampling trough aspiration [12]. Perma curette (Lusofarmaco^®^, Milan, Italy) consists of a spatula with sharp edges at the top of a stick that slides inside a flexible cannula [13]. The histological samples suitable for a diagnosis are obtained by turning the rhomboidal end of the spatula, while the cytological diagnosis is also made possible by connecting the cannula to an instrument that allows to obtain aspiration and/or washing of the endometrial cells [14,15,16].

As enough tissue is needed for an accurate histopathological diagnosis, blind procedures could be underperforming particularly in postmenopausal women. However, similar rates of adequate endometrial biopsy with both blind and hysteroscope-guided endometrial sampling were published [16]. In the literature, it is reported that endometrial sampling without video-guide vs. hysteroscope-guide could be equivalent methods for obtaining sufficient endometrial biopsy [16].

To optimize the waiting list time before the gold standard, and considering the organizational and practical costs [11], it could be helpful to evaluate other ancillary procedures. Prior to the introduction of the hysteroscopic method, curettage was the gold standard for uterine sampling, and subsequently it was partly replaced by simpler and less expensive techniques such as Perma curette, often still used in outpatient hysteroscopies [11].

Nowadays, an endometrial biopsy is typically performed in the clinic to assess women who have abnormal uterine bleeding. An endometrial sample is required for detection due to the rise in the use of hormone medications, or for follow-up to evaluate responsiveness in women who underwent hormone treatment for premalignant endometrial abnormalities [8]. Additionally, the endometrial biopsy can help differentiate between anovulatory and ovulatory bleeding and rule out hyperplastic disease or cancer.

One of the main possible applications could be the use of Perma in patients with alarm symptoms (i.e., genital bleeding in menopause) waiting for hysteroscopy, as for suspicion of endometrial cancer (EC) [8]. Naturally, a negative result would not imply a benign diagnosis, while a positive result in a short time would be another reason to accelerate the gold standard technique and focus on the patient’s health situation.

A blind sample technique could be recommended in situations where an examination under anesthesia is not feasible and/or to reassure the patient or the doctor waiting for hysteroscopic confirm [11]. For these reasons, and the possible excellent prognosis in the early treatment of endometrial tumor, it is essential to favor all possible, easy, rapid, economical, and outpatient diagnostic techniques such as Perma, confirmed by the gold standard.

On this basis, we were interested in the diagnostic accuracy of Perma compared to a hysteroscopic guided sampling analyzing pathological reports. Further, the necessity of having an experienced operator for a successful procedure was evaluated by comparing histological results and their concordance rate among four medical doctors (two experts and two non-experts).

## 2. Materials and Methods

This is a prospective study approved by the Local Ethics Committee of the Azienda Ospedale Università (AOU) Maggiore of Novara, Italy, with the protocol number CE019/2023, 341/CE issued on 21 March 2023.

Women who underwent hysteroscopy from November 2021 to April 2024 at the AOU Maggiore della Carità were asked to participate in this study by accepting to undergo a Perma biopsy along with the hysteroscopic procedure. Patients who agreed to participate were randomly included in a time-consequent way. The samples were from the same patient comparing data obtained with Perma (Figure 1) and those from hysteroscopy (Figure 2).

A hysteroscopy consists of the insertion of a hysteroscope through the cervix and into the uterus, allowing the doctor to visualize the lining of the uterus and the openings of the fallopian tubes through the injection of fluid. It can also be used to take biopsies, remove tissue, or perform other procedures. Perma curette is an endo-sampling that does not require endoscopic guidance or the insertion of liquids: this instrument is introduced into the uterus and rotated without evident discomforts [6,7,13].

Informed consent was obtained for participation in this study from all patients. A gynecological ultrasound check was used for pelvic examination at least one month before hysteroscopy. Endometrial thickness, transverse, longitudinal, and antero-posterior dimensions were reported. Exclusion criteria were (1) deny participating in the study, (2) active genital infections, (3) pathology that could alter the sample (i.e., anogenital fistulas, uterine malformations, etc.), (4) work-up for infertility or recurrent abortion. The surgical team consisted of two doctors, one expert, and one non-expert. The expert medical doctors are professors in the field with more than 10 years of experience; the two non-experts are represented by two residents with less than 5 years of activity. Operators were chosen according to the random order indicated in the surgical schedule by a healthcare professional, unaware of this study, but responsible for drawing up the operating order based on the shift work. After the preparation and disinfection of the surgical field, the non-expert gynecologist first attempts to perform the Perma sampling without any type of cervical dilatation. In case of failure, the expert doctor tries to complete the procedure. Then, a standard hysteroscopy is performed with visual endometrial biopsy. The four operators remained the same for the study period. The samples collected were processed as routine hospital procedures by the pathologists, unaware of this study. Atrophic endometrium, endometrial polyps, and leiomyomas were considered negative, while any other dysplastic pattern of tissue, sufficient for histological diagnosis, was considered a positive finding. The findings were further subdivided into 5 categories: (1) no hyperplasia, (2) simple hyperplasia, (3) complex hyperplasia, (4) atypical hyperplasia, (5) cancer hyperplasia and Endometrial Cancer (EC) [17,18] as to current guidelines.

Then, data were compared within the doctors’ group (inter- and intra-variation) in terms of Cohen’s Kappa. Our results are described in accord with Strobe Guidelines [19].

### Statistical Analysis

Considering previous study suggestions [8,16,20,21] the sample size was calculated based on 13% of tissue samples that were inadequate in hysteroscope guided biopsies and 7% in specimens obtained by endometrial not video-guided techniques. Considering these values as a reference, the minimum required sample size with 7.5% error margin and 5% level of significance is 78 patients. The formula used in this study was *N* ≥ *p* (1 − *p*) (MEZα). Zα is the value of Z at a two-sided alpha error of 5%, ME is the margin [16].

Of the 135 recruitable patients, we successfully performed combined hysteroscopy and Perma sampling in 82 of them. A total of 53 patients were excluded, as 30 refused to participate in this study, and in 23 cases cervical stenosis precluded the insertion of Perma (Figure 3).

Demographic, clinical, and surgical information was collected from clinical records and anonymized with a correspondent identification code in an online database, managed by two authors (C.I.A and F.M.).

The evaluations were made in the same patient comparing data from the samples obtained with Perma and those from hysteroscopy.

The results were first compared within the same patient and subsequent within the doctors’ group (inter- and intra-variation) in terms of Cohen’s Kappa.

The used Cohen’s Kappa formula is as follows: κ = (Po − Pe)/(1 − Pe), where Po is the observed proportion of agreement, and Pe is the proportion of agreement expected by chance.

There are different “degrees of agreement” according to which we can define whether Cohen’s Kappa is poor or excellent, considering these values:-<0, no agreement;-0–0.4, poor agreement;-0.4–0.6, fair;-0.6–0.8, good;-0.8–1, excellent agreement [22].

Sensitivity is considered as the probability of correctly identifying a positive case, specificity as the probability of correctly identifying a negative case [23]. Categorical variables were presented in number and percentage (%) and continuous variables were presented as mean ± SD. The association between categorical variables was evaluated using Fisher’s exact test. The *p* value of <0.05 was considered statistically significant. The statistical analysis was conducted using STATA software, version 17 (StataCorp. 2021. College Station, TX, USA: StataCorp LLC).

## 3. Results

The main characteristics of the studied women are described in Table 1. The patients were predominantly overweight and 55 years old, half of the sample in menopause. There are no statistical associations with body mass index (BMI), parity, or previous intrauterine surgery.

The main reasons for the exam were: suspected polyp, endometrial thickening, and abnormal genital blood loss (Table 2).

Of the eleven endometrial tumors diagnosed at hysteroscopy, nine could be diagnosed by Perma too. Perma showed a sensitivity of 81.8% and a specificity of 100%.

In seventeen cases, invalid samples resulted from the Perma sampling; interestingly in three of them, the hysteroscopic biopsy was considered inadequate for histological judgment by the pathologist. The comparison between valid (both hysteroscopy and Perma) versus non valid was not significant (*p* = 0.583). Menopause significantly increases the probability of an invalid Perma test four times (*p* = 0.03).

Cohen’s Kappa among the non-experts was 0.43 (moderate) while it was 0.3 (fair) among the experts. The highest value recorded was between one non- and one expert 0.68 (substantial) (Table 3).

## 4. Discussion

Prior to the introduction of the hysteroscopic method, curettage was the gold standard for uterine sampling: at that time, simpler and cheaper techniques such as Novak curette, Vabra aspirator, Pipelle catheter, Endo-sampler, and Perma curette were developed in the effort to reach an outpatient diagnostic method [6,10].

In an era of ever-increasing access to hysteroscopic facilities is still there a place for blind endometrial sampling? A non-video-guided sample could be recommended in situations where an examination under anesthesia is not recommendable and/or to give rapid confirmations to the patient or the doctor waiting for hysteroscopy [24]. Moreover, theoretically, it could be possible to imagine the following scenario: (1) absence of hysteroscopic resources, (2) contraindication to a hysteroscopic procedure (age, health problems, etc.), (3) endometrial screening (in a similar approach to the cervical smear) [24].

Hysteroscopy is a surgical technique that requires ever-increasing expertise in the operating context: training is necessary and linked to the operator’s experience, while the use of Perma is considerably simpler and immediate, although possibly less efficient [8,25]. A possible use of non-video-guided sampling is described also as less related to the hysteroscopic discomforts [15]. The rate of complications varies depending on the technique. Numerous extensive nationwide audits have been conducted: Jansen et al. reported a national audit of the Netherlands that looked at over 11,000 diagnostic and 2500 surgical hysteroscopies [26]. According to the audit, operational hysteroscopy had a lot more complications than diagnostic hysteroscopy. During myomectomy resections in particular, 0.2% of surgical hysteroscopies resulted in clinically substantial fluid overload [26]. There were no statistically significant differences between the causes of uterine perforation, despite the fact that it could happen during both diagnostic (incidence 0.13%) and operative hysteroscopy (incidence 0.76%). The majority of perforations (70%) occurred during dilatation for diagnostic hysteroscopy [26]. Patient preparation and hysteroscopic equipment can minimize complications but not eliminate risks [27].

The aim of this work is not to replace the gold standard procedure, but to enhance alternative techniques in the contexts of logistic and time necessities. For example, during the COVID-19 pandemic, the limited possibility of access to surgical rooms and the difficult diagnosis of tumors such as EC has been highlighted: EC diagnoses decreased by 35%, with a double mean waiting time (70.9 days vs. 49.3 in New York, USA) and a worst stage of the disease in surgical treatment [28,29,30]. The use of Perma maybe would have given the possibility of detecting the tumor at an earlier time and more rapidly, with better outcomes in terms of the therapeutic process.

In high-income nations, endometrial carcinoma is the most prevalent gynecological cancer, and its widespread distribution could be influenced by several factors [31,32]. Because the disease manifests itself quite early in its course by irregular vaginal bleeding, 80% of all cases are found in an early phase. With a 5-year overall survival (OS) rate of 90% to 95%, it yields a favorable prognosis [33]. For this reason, and the possible excellent outcomes in the early treatment of this tumor, it is essential to favor all possible, rapid, economical, and outpatient diagnostic techniques [2,3,8]. In the context of EC, the possible benefit of Perma is the easier use: hysteroscopy is a surgical technique that requires ever-increasing expertise in the operating context [2,3]. Training is necessary and linked to the operators’ experience, while the use of Perma is considerably simpler and immediate, although it could be less efficient [25,34].

Perma, compared to hysteroscopy, appeared to be simpler, faster, requires less training, might cause less discomfort, and is cheaper [25,34].

Despite the possible benefits, the literature on the use of Perma is scarce or not updated. Of the few studies present, with a very limited sample, the results are controversial as mainly related to non-video guided endometrial sampling techniques than Perma [35]. A review of last decades affirmed that hysteroscopic guided sample and uterine curettage can have a high risk of underestimation of EC, and this rate could cause inappropriate surgical procedure: hysteroscopic resection seems a possible factor to lower this risk [35]. Instead, comparing biopsy (HS), dilation and curettage (D&C), and Perma, the frozen sections diagnosed EC in 24/36 (66%), 7/16 (43.8%) and 6/14 (42.9%) cases when preoperative exams were performed with HS, D&C and Perma, respectively (*p* = 0.05), and the risk of underlying EC was significantly smaller with D&C or with Perma [35].

In a previous Italian study, the concordance between histological and cytological examinations was described as 50% in endometrial hyperplasia and 100% in early cancers, comparing hysteroscopy and Perma [36]. Other data highlighted that the blind technique could have diagnostic preoperative validity [37].

The subjectivity of the diagnosis prior to histological confirmation is one of the main disadvantages of sampling diagnosed diseases [38]. Despite the great modernity that we are facing thanks to Artificial Intelligence, machine learning, and deep learning models in the use of hysteroscopy, Perma could have a role in the context of immediacy and less required training [38]. For these reasons, we were also interested in analyzing the results of this diagnostic tool performed in the hands of both expert and non-expert medical doctors, and their concordance of opinions about the histological results after Perma vs. Hysteroscopy. Perma has a sensitivity of 81.8% and a specificity of 100% compared to hysteroscopy (the gold standard), with a good operator in accordance in our study. Comparing scientific data and the attitude of the operators is fundamental: there is no univocal scientific intention about hysteroscopic training, despite the fact that studies have evaluated several factors, such as the use of simulators: Perma could partially solve this complexity [34]. This is an interesting consideration in favor to Perma use, but it needs to be improved with further studies.

In addition to clinical outcomes, we analyze the agreement between gynecologists on histological examinations. The evaluation of the results of the two techniques led to a good agreement between our operators, different experts in the gynecological field: Cohen’s Kappa between residents was 0.43 (moderate), between professors was 0.30 (fair), the highest concordance was reported between one resident and one professor and was 0.68 (substantial).

In the future, it could also be helpful to understand how some other conditions than doctors’ experience, such as menopause, impact on endometrial sample techniques. As for our results, menopause is a known limiting condition described in the literature, probably linked to atrophy and other related factors [39,40].

### Limitations

More data in the literature are related to blind endometrial sampling techniques and not only on Perma. Our sample needs to be increased. Possible bias could be related to patient selection and diagnostic verification. Moreover, it was impossible to blind the surgeon and pathologist about the technique to use for endometrial sampling. Experts’ and non-experts’ opinions about the histological diagnosis could be influenced by their different working experience.

## 5. Conclusions

The literature has solidly established the overwhelming reasons in favor of an office-based endometrial biopsy accuracy, patient and practitioner convenience, and cost containment. Outpatient screening practices will remain crucial to the diagnostic abilities.

Perma represents a pragmatic diagnostic tool, which can also be used outpatient or in the emergency room to obtain a rapid initial diagnosis and define the subsequent therapeutic process.

## Figures and Tables

**Figure 1 biomedicines-13-01113-f001:**
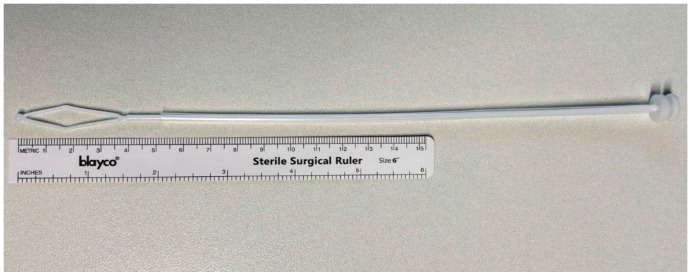
Perma curette (Lusofarmaco^®^). (AOU Maggiore della Carità, Novara, Italy).

**Figure 2 biomedicines-13-01113-f002:**
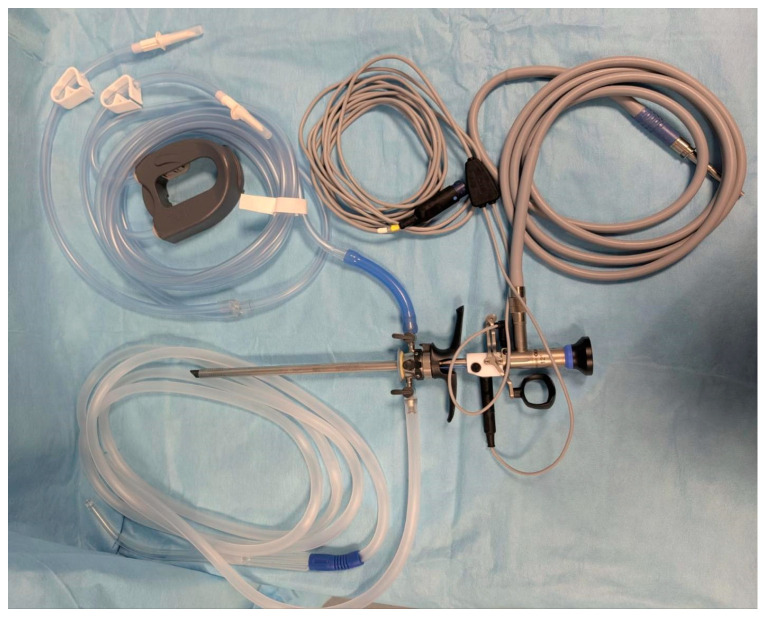
Hysteroscopy (Olympus^®^). (AOU Maggiore della Carità, Novara, Italy).

**Figure 3 biomedicines-13-01113-f003:**
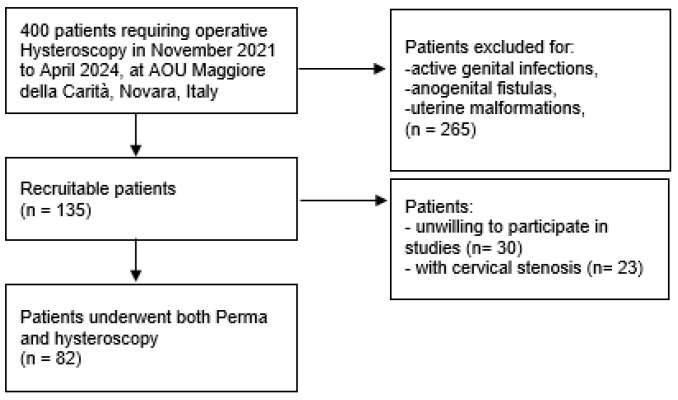
Recruitment sample flow chart.

**Table 1 biomedicines-13-01113-t001:** Sample characteristics.

	Mean±	SD
Age (years)	55.5	11.3
Weight (kg) *	70.8	16.2
Height (cm) *	162.3	7.3
BMI * (kg/m^2^)	26.7	5.0
Menopause (years)	50.6	3.4
	**Yes**	**No**
Smoke	69/82 84.1%	13/82 15.8%
Hypertension	56/82 68.3%	26/82 31.7%
Diabetes	74/82 90.2%	8/82 9.8%
Tumors	72/82 87.8%	10/8 12.2%
Parity *	G0 30.5% G1 24.4% G2 32.9% G3 9.8% G4 2.4%	

* BMI = body mass index; kg for kilograms; cm for centimeters; G for the parity.

**Table 2 biomedicines-13-01113-t002:** Suspected diagnosis prior to the exam.

	N	%
Endometrial polyp	32	39.0
Endometrial thickening	31	37.8
Endometrial hyperplasia, dysplasia	4	4.9
Genital bleeding	19	23.2
Myoma	8	9.8
Adhesions	1	1.2
Cervical polyp	3	3.7
Other	4	4.9

More than one option possible for each patient.

**Table 3 biomedicines-13-01113-t003:** Concordance rate.

Perma	Tumor 10.9% vs. Negative 89.1%
Hysteroscopy	Tumor 13.4% vs. Negative 86.6%
Percentage of outcome concordance—Expert 1	52/82 (63.4%)
Percentage of outcome concordance—Expert 2	50/82 (60.9%)
Percentage of outcome concordance—Non-expert 1	58/82 (70.7%)
Percentage of outcome concordance—Non-expert 2	72/82 (87.8%)
Percentage of outcome concordance between Experts	A: 65.8% EA: 52.9% K = 0.2743
Percentage of outcome concordance between Non-experts	A: 80.5% EA: 65.7% K = 0.4315
Expert 1 and Non-expert 1	A: 65.8% EA: 55.6% K = 0.2316
Expert 1 and Non-expert 2	A: 63.4% EA: 60.1% K = 0.0821
Expert 2 and Non-expert 1	A: 85.4% EA: 54.5% K = 0.6780
Expert 2 and Non-expert 2	A: 70.7% EA: 58.3% K = 0.2981

A = agreement; EA = expected agreement; K = Cohen’s Kappa.

## Data Availability

Data will be made available on request.

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
