# Peer review of "Perma Curette and Hysteroscopy: An Observational Study About Endometrial Sampling"

_biomedicines, 2025, doi:10.3390/biomedicines13051113_

Round 1

Reviewer 1 Report

Comments and Suggestions for Authors

Dear authors;

Thank you for your fluent writing. Below I provide my commentaries.

Title:

  • Based on your result and discussion, I think it is better to rewrite your title and show the comparison between Perma and hysteroscopy.

Abstract:

  • Please add the statistical analysis in the end of M&M section.
  • Write the P value in correct way in line 23 “p 0.583”
  • What is the unit of “0.43” and “0.30” in line 26.
  • Please add a conclusion in the end of abstract.

Introduction:

  • Please add some references for line 31-34.
  • As we can see in line 38, endo-sampler has a better sensitivity and specificity in compared to Perma. It is better to add more explanation in introduction or even address it in Discussion.
  • Use the figure 1 for M&M section.
  • Please write better about Perma curette application.

Materials and Methods:

  • Can you add a clinical picture of sampling or compared the samples in two methods?
  • Line 132, mean±SD instead of “mean and SD”

Results:

  • Line 148, is there any significate difference between Perma and hysteroscopy? Please add p value and mean±
  • Please write the p value in italic form in line 152 and 153.
  • Line 154, add the unit of “0.43” and “0.30”.
  • can you add any figure or chart in your manuscript?

Discussion:

  • Line 160-172, it is introduction. Please move it there. Instead, explain your aim and results just in first paragraph.
  • Line 173-181, where are your references? In addition, there are not discussion.
  • I need to read a stronger discussion and highlight the importance of using Perma or its limitations. You just mentioned some tips in lines 182 and 197-198.

Author Response

Thank you for your fluent writing. Below I provide my commentaries.

-Thanks for your appreciation and precious commentaries. Every requested changes is highlighted in the text in track changes.

Title:

  • Based on your result and discussion, I think it is better to rewrite your title and show the comparison between Perma and hysteroscopy.

-Thanks, we changed as suggested.

Abstract:

  • Please add the statistical analysis in the end of M&M section.
  • Write the P value in correct way in line 23 “p 0.583”
  • What is the unit of “0.43” and “0.30” in line 26.
  • Please add a conclusion in the end of abstract.

-Thanks, we modified these points as requested.

As described also in the text, Cohen's Kappa is a measure of agreement between two raters or between the same rater at different times, when the ratings are qualitative or categorical. Kappa values ​​range from -1 to +1, with positive values ​​indicating agreement and negative values ​​indicating disagreement. A Kappa of 1 indicates perfect agreement, while a Kappa of 0 indicates agreement equal to that expected by chance.

Introduction:

  • Please add some references for line 31-34.

-Thanks, we added as requested.

  • As we can see in line 38, endo-sampler has a better sensitivity and specificity in compared to Perma. It is better to add more explanation in introduction or even address it in Discussion.

-Thanks, we added as requested. The endosampler is a sampling method with good applicability and well described in the literature. Our intent was to verify the applicability of Perma given the little information described so far.

  • Use the figure 1 for M&M section.

-We moved it, thanks.

  • Please write better about Perma curette application.

-We added as requested.

Materials and Methods:

  • Can you add a clinical picture of sampling or compared the samples in two methods?

-We added textual explanation.

  • Line 132, mean±SD instead of “mean and SD”

-Thanks, we corrected.

Results:

  • Line 148, is there any significate difference between Perma and hysteroscopy? Please add p value and mean±

-Significative p-values are listed, we modified with mean±.

  • Please write the p value in italic form in line 152 and 153.
  • We modified, thanks.
  • Line 154, add the unit of “0.43” and “0.30”.

- As explained, Cohen's Kappa is a measure of agreement between two raters or between the same rater at different times, when the ratings are qualitative or categorical. Kappa values range from -1 to +1, with positive values indicating agreement and negative values indicating disagreement. A Kappa of 1 indicates perfect agreement, while a Kappa of 0 indicates agreement equal to that expected by chance.

  • can you add any figure or chart in your manuscript?

-We added fig.2, as requested.

Discussion:

  • Line 160-172, it is introduction. Please move it there. Instead, explain your aim and results just in first paragraph.
  • Line 173-181, where are your references? In addition, there are not discussion.
  • I need to read a stronger discussion and highlight the importance of using Perma or its limitations. You just mentioned some tips in lines 182 and 197-198.

-Thanks, we modified as requested.

Reviewer 2 Report

Comments and Suggestions for Authors

The manuscript addresses an important clinical question regarding the diagnostic validity of the Perma curette compared with hysteroscopically guided biopsy. However, several aspects require improvement:

Methods:

Clarify the randomization process for choosing between expert and non-expert operators to improve transparency and reproducibility.

Expand the statistical methodology section, specifically explaining how sensitivity, specificity, and Cohen’s Kappa were calculated.

Results:

Include confidence intervals (CIs) for sensitivity and specificity to allow readers to understand the precision of these estimates.

The discussion of inter-rater reliability using Cohen’s Kappa could benefit from further clarification, specifically detailing why agreement might differ between expert and non-expert operators.

Public Health Relevance:

Strengthen the discussion of the public health implications of widespread adoption of the Perma Curette, focusing on accessibility, cost-effectiveness, and patient comfort compared to standard hysteroscopic techniques.

Clearly discuss potential implications for clinical practice guidelines or recommendations.

Limitations:

Expand on the identified limitations, particularly by addressing potential biases related to patient selection and diagnostic verification.

Author Response

The manuscript addresses an important clinical question regarding the diagnostic validity of the Perma curette compared with hysteroscopically guided biopsy. However, several aspects require improvement:

-Thanks for your precious suggestions, we modified the text in track changes.

Methods:

Clarify the randomization process for choosing between expert and non-expert operators to improve transparency and reproducibility.

Expand the statistical methodology section, specifically explaining how sensitivity, specificity, and Cohen’s Kappa were calculated.

-Thanks, we added as requested.

Results:

Include confidence intervals (CIs) for sensitivity and specificity to allow readers to understand the precision of these estimates.

-Thanks. We list main intervals:

Perma vs Hysteroscopy 0.025 (95% CI 1.187871 to 13.7027) 0.00 (95% CI .0395489 to .312162)

Uterine surgery (i.e. uterine revision) 0.468 (95% CI .5091743 to 4.340362) 0.000      (95% CI .100693 to .4642771)

Diagnosis of cancer trough perma vs hysteroscopy 0.546 (95% CI .1669731 to 2.579242) 0.000   (95% CI  .1577535  to  .5174698)

The discussion of inter-rater reliability using Cohen’s Kappa could benefit from further clarification, specifically detailing why agreement might differ between expert and non-expert operators.

-Thanks for the suggestion, we stress the concept in discussion and limitation.

Public Health Relevance:

Strengthen the discussion of the public health implications of widespread adoption of the Perma Curette, focusing on accessibility, cost-effectiveness, and patient comfort compared to standard hysteroscopic techniques.

Clearly discuss potential implications for clinical practice guidelines or recommendations.

-Thanks, we modified as requested (introduction and discussion).

Limitations:

Expand on the identified limitations, particularly by addressing potential biases related to patient selection and diagnostic verification.

-Thanks, we modified as requested.